# The prevalence of diabetes amongst young Emirati female adults in the United Arab Emirates: A cross-sectional study

Maysm N. Mohamad[1], Leila Cheikh Ismail[2,3], Lily Stojanovska[1,4], Vasso Apostolopoulos[4], Jack Feehan[4,5], Amjad H. Jarrar[1], Ayesha S. Al Dhaheri[1]*

1 Department of Nutrition and Health, College of Medicine and Health Sciences, United Arab Emirates University, Al Ain, United Arab Emirates, 2 Clinical Nutrition and Dietetics Department, College of Health Sciences, University of Sharjah, Sharjah, United Arab Emirates, 3 Nuffield Department of Women's & Reproductive Health, University of Oxford, Oxford, United Kingdom, 4 Institute for Health and Sport, Victoria University, Melbourne, VIC, Australia, 5 Department of Medicine–Western Health, The University of Melbourne, Melbourne, VIC, Australia

* ayesha_aldhaheri@uaeu.ac.ae

## Abstract

### Aims

The prevalence of type 2 diabetes is rapidly increasing in the United Arab Emirates (UAE). The purpose of this study was to investigate the prevalence of prediabetes and diabetes using FPG and HbA1c and to examine their relationships with obesity and other risk factors in young female Emirati college students.

### Methods

In this cross-sectional study we recruited 555 female college students aged 17–25, enrolled at United Arab Emirates University in Al Ain, UAE. Anthropometric analysis, blood pressure, and various biochemical markers were measured using standard methods. Type 2 Diabetes, impaired fasting plasma glucose (FPG), and elevated HbA1c levels were examined in the study population as per the standards of medical care in diabetes, set out by the American Diabetes Association in 2020.

### Results

Based on the HbA1c test, the prevalence of pre-diabetes and diabetes were 24% and 8.6%, respectively. Based on the FPG test, the prevalence of pre-diabetes and diabetes were 9.2% and 0.5%, respectively. The kappa statistic of concordance between HbA1c and FPG was 0.287, $P < 0.001$. Abnormal glycemic status was significantly associated with decreased high-density lipoprotein (HDL) level (< 50 mg/dl) ($p = 0.002$) and elevated high-sensitivity C-reactive protein (Hs-CRP) level ($\geq 2.0$ mg/L) ($P < 0.001$).

### Conclusions

Using FPG to evaluate glycemic control seems to underestimate the burden of undiagnosed diabetes which could have a significant impact on clinical practice. Our data indicates an

**Data Availability Statement:** All data files are available at https://doi.org/10.6084/m9.figshare.14547246.v1.

**Funding:** This research was funded by United Arab Emirates University, grant number CFA-31F038. A. A.D received all funding for this study. The funders had no role in study design, data collection and analysis, decision to publish, or preparation of the manuscript.

association between abnormal glycemic status with HDL and Hs-CRP. Further evaluation is needed to assess the impact of using HbA1c as a diagnostic test for diabetes in the UAE.

# 1. Introduction

Type 2 diabetes (T2D) is a growing concern for healthcare systems globally, commonly referred to as having reached pandemic status. In recent decades the global incidence of T2D has grown significantly, from 4.3% to 9.6% in men, and 5% to 9% in women [1, 2]. The negative health outcomes of T2D are largely mediated through an increased risk of cardiovascular disease. T2D is a major risk factor for a wide range of cardiovascular conditions, including coronary heart disease and myocardial infarction, stroke, and peripheral vascular disease [3]. This is further compounded by the intersection of several other common risk factors in people living with T2D, such as obesity, physical inactivity, chronic inflammation, and hyperlipidemia. Once in place, T2D is challenging to reverse, making early identification and prevention essential. The term prediabetes refers to a state of impaired insulin signaling and hyperglycemia that has not yet reached the threshold for diagnosis of diabetes [4]. Prediabetes is an important indicator, with as many as 70% of those diagnosed with it, eventually going on to become diabetic [4]. As with diabetes, prediabetes is associated with a number of other cardiovascular risk factors, likely due to shared mechanisms and etiologies.

T2D is a pathological metabolic condition resulting from chronic glucose exposure. This leads to a state in which the body has become resistant to insulin signaling, with a resulting accumulation of glucose in the circulation, known as hyperglycemia. Hyperglycemia is diagnostic of T2D, however, there are several accepted methods of its assessment, with the two most common being, the fasting plasma glucose (FPG) test or evaluation of glycated hemoglobin level (HbA1c). FPG evaluates the level of glucose remaining in the circulation after fasting for at least eight hours, while HbA1c allows for the assessment of red blood cell exposure to blood glucose over the previous three months. The American Diabetes Association classifies someone as prediabetic with an FPG level of 100-125mg/dl [5.6–6.9 mmol/L], or/and HbA1c of 5.7–6.4% [39-47mmol/mol], and as diabetic when the upper boundary of prediabetes is exceeded for either test [5]. These tests are particularly useful for screening, especially in large populations as they require only a single blood test, in comparison to the oral glucose tolerance test (OGTT) which requires multiple tests over a 2h period. In 2009, the international expert committee recommended that HbA1c become the standard criteria for the diagnosis of diabetes [6], with the World Health Organization (WHO) accepting its use in 2011 [7].

While the growth in incidence globally is concerning, there is significant regional variation in prevalence, with many regions exceeding this average growth, particularly in countries that have undergone rapid westernization of diet and activity such as the United Arab Emirates (UAE) [8]. In 2010, the UAE had the prevalence of T2D with 18.7% of the population living with the disease, second only to Nauru (30.9%), and it is predicted to reach 21.4% by the year 2030 [9]. In 2019, the international diabetes federation (IDF) reported a nationwide prevalence of 15.4% in adults (20–79) [1], however in the major metropolitan centers where the majority of the population live the numbers are higher. In the more densely populated northern emirates, 25.1% of the population was found to have diabetes in 2013 [10]. In the capital of Abu Dhabi, it was found that 27.1% of the population were prediabetes [11]. These high rates are particularly of concern in the face of the cardiovascular disease burden associated with T2D. Chronic cardiovascular disease is the largest cause of mortality in the UAE, responsible for 40% of all deaths in 2016 [12]. As a key risk factor of cardiovascular disease, this places T2D as a critical health issue for the UAE in the future.

University students are transitioning form adolescence to adulthood. During this period, their dietary and health behaviors are known to change, and they are often forming long-lasting detrimental lifestyle behaviors [13]. Therefore, focusing on young adults to identify opportunities for early intervention is essential, as there is scope to prevent the transition from prediabetic to diabetic, and potentially reverse the early stage of T2D. Moreover, there is no previous research to date identifying risk factors of altered glucose metabolism using HbA1c and FPG among female young adults.

The purpose of this study was to determine the prevalence of prediabetes and T2D among a sample of young Emirati female adults and to examine its association with other risk factors. This study also aims to examine the concordance between HbA1c and FPG in their capacity to diagnose prediabetes and T2D, as both tests are widely used throughout the UAE.

## 2. Methods

### 2.1 Study population

Emirati female students attending United Arab Emirates University (UAEU) were enrolled in this cross-sectional study between January 2014 and May 2016. Recruitment was carried out via email invitation. Interested students were asked to complete a health-screening questionnaire before taking part in the study to confirm that they met the inclusion criteria. Participants were excluded if they did not show for the appointment, refused or failed blood collection, were pregnant, breastfeeding, or were on long-term medication. Participants were asked to read the information sheet carefully and were given the chance to ask questions. All participants provided a written informed consent before taking part in the study. Informed consent was obtained from parents or guardians for individuals under the age of 18. The study was conducted according to the ethical principles of the Declaration of Helsinki, and ethical approval was obtained from the UAEU Scientific Research Ethics Committee (Reference number DVCRGS/370/2014).

### 2.2 Measures

A self-report questionnaire was used to record the family medical history of non-communicable diseases as well as demographic information. Anthropometric measurements were conducted in the Department of Nutrition and Health laboratory at the UAEU. All measurements were obtained after a 12 hour fast while wearing minimal clothing (within local cultural limits) and no shoes. To minimize inter-day fluctuations all biochemical measurements were taken between 7:00–10:00 am. Participants were asked not to attend for assessment during their menstrual period. Participants included in this study have never and do not consume alcohol for cultural and religious reasons.

Height was measured in a standing position via a stadiometer (Seca Stadiometer, Seca Ltd, Birmingham, UK) to the nearest millimeter. Body composition analysis, as well as weight, were assessed via a Tanita segmental body composition analyzer (Tanita BC-418, Tanita Corp., Tokyo, Japan). Body mass index (BMI) was calculated by dividing the weight in kilograms by the height squared in meters ($kg/m^2$). Assessment of skinfold thickness was used to measure total body fat at four sites (biceps, triceps, subscapular, and suprailiac) with total fat estimation calculated via the equation described by Durnin and Womersley [14]. To measure waist circumference a plastic tape was used at the midway point of the inferior margin of the ribs and superior border of the iliac crest to give a result in centimeters. In obese participants, this was measured at the level of the umbilicus [15]. Hip circumference was measured similarly, at the level of the posterior most point of the buttocks [15]. These measurements were

used to calculate the Waist-Hip ratio (WHR) as waist circumference divided by the hip circumference.

Systolic and diastolic blood pressures were measured by a registered nurse using a digital automated sphygmomanometer (Omron Hem-907, Omron Healthcare, Kyoto, Japan), after 15 minutes of seated rest. The instrument was validated before the study and calibrated regularly. The reading was repeated after 5 minutes and the average of the 2 measurements was recorded. A registered nurse performed a peripheral venipuncture and collected the blood samples into sterile EDTA vacutainer tubes from each participant after a 12 hour fast. Whole blood was immediately used to assess hemoglobin (Hb) concentration and HbA1c percentage by HemoCue Hb 201 and portable photometer system (HemoCue AB, Sweden), respectively. Blood samples were aliquoted and distributed into serum separator tubes with clot activator (Vacutest Kima, Italy), centrifuged at 2,500 rpm, for 15 min, and the resulting serum stored at -80˚C until time of analysis. Total cholesterol (TC), triglyceride (TG), LDL-cholesterol, high-density lipoprotein-cholesterol (HDL-C), high sensitivity C-reactive protein (hs-CRP), and FPG in serum were analyzed using Cobas C111 automated biochemical analyzer (Roche Diagnostics, Indianapolis, IN, USA).

### 2.3 Definitions

In accordance with the ADA (2020) criteria, T2D was determined as FPG ≥ 126 mg/dL and/or HbA1c ≥ 6.5% (48 mmol/mol). Prediabetes was defined as an FPG levels between 100 and 125 mg/dL and/or elevated HbA1C range (5.7–6.4% [39–47 mmol/mol]) [5]. Cut-off points for body BMI, fat percentage, WHR, and anemia were based on the WHO recommended values [15, 16]. Central obesity was defined as waist circumference ≥ 80 cm [15]. The American College of Cardiology and the American Heart Association guidelines were used to define hypertension. Elevated blood pressure was defined as systolic blood pressure 120-129mmHg and diastolic blood pressure < 80 mmHg. Hypertension stage 1 was defined as systolic blood pressure 130-139mmHg or diastolic blood pressure 80–89 mmHg. Hypertension stage 2 was defined as systolic blood pressure ≥ 140 mmHg and/or diastolic blood pressure ≥ 90 mmHg. Anemia was defined as hemoglobin < 120 g/L [16]. Abnormal lipid profile cutoffs were based on the 2019 American College of Cardiology and the American Heart Association guidelines as ≥150 mg/dL triglyceride, ≥150 mg/dL total cholesterol, > 100mg/dL LDL-C and < 50mg/dL HDL-C, and elevated HsCRP as ≥ 2 mg/L.

### 2.4 Statistical analyses

Data analyses were carried out using SPSS software, version 26.0 (SPSS, Chicago, IL, USA). Categorical data were expressed as counts and percentages. Cohen's K was used to determine if there was an agreement between FPG and HbA1c in diagnosing diabetes and prediabetes among the studied population. Univariate and multivariate logistic regressions were used to study the association between age, family history of diabetes or hypertension, blood Pressure, body mass index, body fat, waist circumference, waist-hip ratio, hemoglobin, triglyceride, total Cholesterol, LDL, HDL, High-Sensitivity C-Reactive Protein with abnormal glycemic status as the outcome variable. *P* values < 0.05 were considered statistically significant.

## 3. Results

### 3.1 Sample characteristics

Of 885 students who received the invitation to participate, 654 showed an interest, giving a response rate of 74%. Of these, 555 participants took part in this study, with 99 excluded. Fig 1

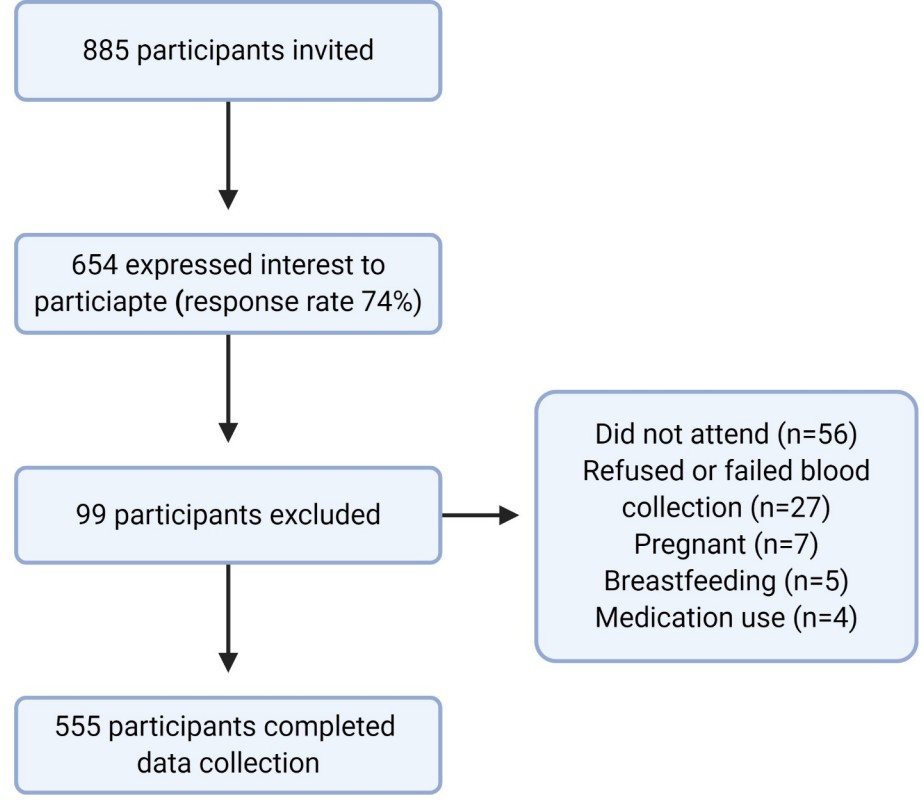

**Fig 1. Flow chart of study inclusion.**

shows the recruitment, inclusion, and exclusion data for the study. The participants were aged between 17 and 25, with a mean age of 20.4 (± 1.7 years). Table 1 describes the demographic and clinical characteristics of the participants. The majority of the participants (55.0%) had at least one parent who had diabetes or was hypertensive. More than one third of the study sample were classified as overweight or obese (23.1% and 10.5%, respectively), based on their BMI. Additionally, central obesity was prevalent among 18.2% of the studied population based on the WHO waist circumference classification, and 8.3% had a WHR $\geq$ 0.8. While none of the participants reported as being previously diagnosed with elevated blood pressure, 11.3% were found to have stage 1 or stage 2 hypertension. Most of the participants had triglyceride levels < 150 mg/dl (98.6%), while 55.9% had a total cholesterol level > 150 mg/dL and 31.2% had LDL levels > 100 mg/dL. Low HDL (< 50mg/dL) and hemoglobin (< 120 g/L) levels were also common among participants (48.8% and 51.2%, respectively). Hs-CRP was $\geq$ 2.0 mg/L in 17.5% of participants.

### 3.2 Prediabetes and diabetes are prevalent in young Emirati adult females

Using HbA1c as a criterion, the prevalence of prediabetes in the sample was 24% (95% CI: 20.4 to 27.5, n = 133) and the prevalence of diabetes was 8.6% (95% CI: 6.3 to 11.0, n = 48). However, when assessed via the criterion of FPG level, the prevalence was lower, with 9.2% prediabetic (95% CI: 6.8 to 11.6, n = 51), and 0.5% diabetic (95% CI: 0.1 to 1.2, n = 3). With both tests combined, the prevalence of prediabetes was 24.7% (95% CI: 21.1 to 28.3, n = 137) and the prevalence of diabetes was 8.6% (95% CI: 6.3 to 10.9, n = 48) (Fig 2).

**Table 1. Demographic and clinical characteristics of female participants (n = 555), at United Arab Emirates University (UAEU), Al Ain, UAE.**

| Characteristics | Mean ± SD |
|---|---|
| Age (years) | 20.4 ± 1.7 |
| Body mass index (kg/m$^2$) | 23.8 ± 5.1 |
| Waist circumference (cm) | 71.4 ± 10.0 |
| Waist-hip ratio | 0.7 ± 0.05 |
| Body fat (%) | 29.6 ± 9.0 |
| Triglyceride (mg/dL) | 60.9 ± 24.9 |
| Total cholesterol (mg/dL) | 156.3 ± 32.4 |
| Low Density Lipoprotein (mg/dL) | 91.7 ± 25.3 |
| High Density Lipoprotein (mg/dL) | 52.2 ±16.4 |
| Hemoglobin (g/L) | 117.9 ±14.7 |
| Fasting plasma glucose (mg/dL) | 83 ± 12.2 |
| Glycated hemoglobin (%) | 5.6 ± 0.9 |
| **Age (years), n (%)** | |
| 17–19 | 194 (35.0) |
| 20–22 | 299 (53.9) |
| 23–25 | 62 (11.2) |
| **Parents with diabetes or hypertension, n (%)** | |
| Yes | 305 (55.0) |
| No | 250 (45.0) |
| **Body mass index (kg/m$^2$), n (%)** | |
| Underweight ($<$ 18.5) | 62 (11.2) |
| Normal weight (18.5–24.99) | 307 (55.3) |
| Overweight (25–29.99) | 128 (23.1) |
| Obese ($>$ 30) | 58 (10.5) |
| **Waist circumference (cm), n (%)** | |
| $<$ 80 | 454 (81.8) |
| $\geq$ 80 | 101 (18.2) |
| **Waist-hip ratio, n (%)** | |
| $<$ 0.8 | 509 (91.7) |
| $\geq$ 0.8 | 46 (8.3) |
| **Body fat (%), n (%)** | |
| $<$ 35 | 398 (71.7) |
| $\geq$ 35 | 157 (28.3) |
| **Blood pressure (BP) (mm Hg), n (%)** | |
| Normal ($<$ 120/80) | 453 (81.6) |
| Elevated (systolic 120–129 and diastolic $<$ 80) | 39 (7.0) |
| High BP Stage 1 (systolic 130–139 or diastolic 80–89) | 54 (9.7) |
| High BP Stage 2 (systolic $>$ 140 or diastolic $\geq$ 90) | 9 (1.6) |
| **Triglyceride (mg/dL), n (%)** | |
| $<$ 150 | 547 (98.6) |
| $\geq$ 150 | 8 (1.4) |
| **Total cholesterol (mg/dL), n (%)** | |
| $\leq$ 150 | (44.1) |
| $>$150 | 310 (55.9) |
| **Low Density Lipoprotein (mg/dL), n (%)** | |
| $\leq$ 100 | (68.8) |

*(Continued)*

**Table 1.** (Continued)

| Characteristics | Mean ± SD |
|---|---|
| >100 | 173 (31.2) |
| **High Density Lipoprotein (mg/dL), n (%)** | |
| ≥ 50 | 284 (51.2) |
| < 50 | 271 (48.8) |
| **High-sensitivity C-reactive protein (mg/L), n (%)** | |
| < 2.0 | 458 (82.5) |
| ≥ 2.0 | 97 (17.5) |
| **Hemoglobin (g/L), n (%)** | |
| ≥ 120 | 271 (48.8) |
| < 120 | 284 (51.2) |
| **Fasting plasma glucose (mg/dL), n (%)** | |
| < 100 | 501 (90.3) |
| 100–125 | 51 (9.2) |
| ≥ 126 | 3 (0.5) |
| **Glycated hemoglobin (%), [nmol/mol] n (%)** | |
| < 5.7 [39] | 374 (67.4) |
| 5.7–6.4 [39–46] | 133 (24.0) |
| ≥ 6.5 [48] | 48 (8.6) |
| **Diabetes diagnosis using HbA1c and Fasting plasma glucose criteria, n (%)** | |
| Normal | 370 (66.7) |
| Prediabetes | 137 (24.7) |
| Diabetes | 48 (8.6) |

### 3.3 Discordance in diagnostic measures of glucose metabolism

Cohen's K was used to determine whether there was an agreement between FPG and HbA1c in diagnosing diabetes and prediabetes among the studied population (Table 2). Agreement between the two diagnostic methods was imperfect, with 'fair' agreement via Cohen's K = 0.287 (95% CI, 0.387 to 0.188), P < 0.001 (Table 2). It is worth noting that while the standard

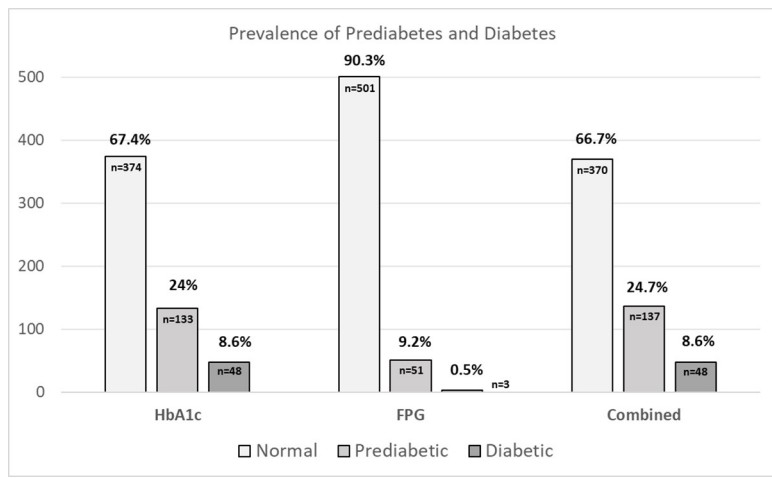

**Fig 2. Prevalence of prediabetes and diabetes based on diagnostic measures.** FPG: Fasting plasma glucose, HbA1c: Glycated hemoglobin.

**Table 2. Agreement between categories of diabetes and prediabetes based on fasting plasma glucose (FPG) and glycated hemoglobin (HbA1c) criteria among young female adults aged 17 to 25 years (n = 555) at UAEU.**

| | | Diagnosis based on HbA1c | | | Total (FPG) |
|---|---|---|---|---|---|
| | | *Normal* | *Prediabetes* | *Diabetes* | |
| **Diagnosis based on FPG** | *Normal* | 370 (66.7) | 97 (17.5) | 34 (6.1) | 501 |
| | *Prediabetes* | 4 (0.7) | 36 (6.5) | 11 (2.0) | 51 |
| | *Diabetes* | 0 (0) | 0 (0) | 3 (0.5) | 3 |
| **Total (HbA1c)** | | 374 | 133 | 48 | 555 |
| **Kappa (95% CI)** | | 0.287 (0.387–0.188) | | | |

Brackets show percentage agreement between measures.

assessment of agreement is 'fair' this is commonly noted as being unacceptable in health research, due to low accuracy [17]. The percentage of participants who were diagnosed with prediabetes and diabetes using HbA1c and FPG are shown in Fig 3. Among 24% of participants having prediabetes based on HbA1c, only 6.5% were classified as having the same diagnosis by FPG (out of 9.2%). Among 8.6% of individuals classified as having diabetes by HbA1c, only 0.5% were classified the same when using FPG.

### 3.4 Risk factors for diabetes and cardiovascular disease

In univariate analyses, older participants (23–25 years) had a significantly higher risk of having abnormal glycemic status (odds ratio [OR]: 1.871; 95% CI: 1.044 to 3.352) than younger participants (17–19 years), however, this effect was not significant in the multivariate analyses (Adjusted odds ratio [aOR]: 1.780; 95% CI: 0.945 to 3.351). Interestingly, neither a family history of diabetes or hypertension, nor elevated blood pressure had a significant association with abnormal glycemic status ($P$ = 0.581 and $P$ = 0.444, respectively) (Table 3).

Participants living with obesity were almost 2.3 times (OR: 2.301; 95% CI: 1.302 to 4.066) more likely to have abnormal glycemic status than those with 'normal' weight ($P$ = 0.004), although this effect was no longer significant in the multivariable analyses (adjusted odds ratio [aOR]: 0.694; 95% CI: 0.243 to 1.985; $P$ = 0.495). Similarly, body fat percentage $\geq$ 35% and waist circumference $\geq$ 80 cm conferred an association with the risk of abnormal glycemic status in univariable analyses (odds ratio [OR]: 1.980; 95% CI: 1.351 to 2.901 and [OR]: 2.561; 95% CI: 1.650, 3.975), respectively; $P$ < 0.001), yet this association was not significant after adjusting for other factors (adjusted OR [aOR]: 1.266, 95% CI: 0.647 to 2.476, and [aOR]: 1.608; 95% CI: 0.692 to 3.737). Furthermore, low HDL level (< 50 mg/dl) and elevated high

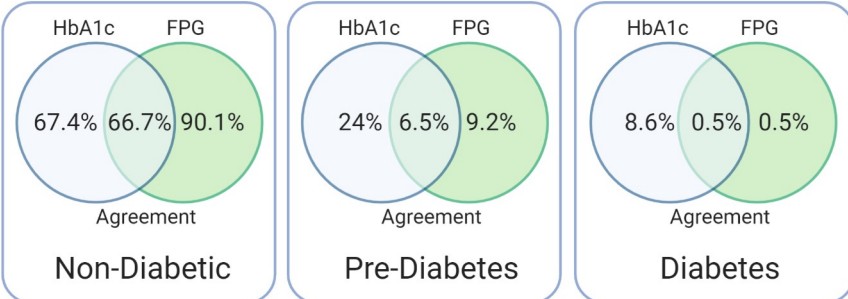

**Fig 3. Diagnostic agreement between fasting plasma glucose and glycated hemoglobin.** FPG: Fasting plasma glucose, HbA1c: Glycated hemoglobin.

**Table 3. Risk factors for abnormal glycemic status based on FPG and HbA1c.**

| Variable | n | With abnormal glycemic status | | | | |
|---|---|---|---|---|---|---|
| | | n (%) | Crude Odds Ratio (95% CI) | P-value | Adjusted Odds Ratio (95%CI) | *P*-value |
| **Age (Years)** | | | | | | |
| 17–19 | 194 | 62 (32.0) | Reference | | Reference | |
| 20–22 | 299 | 94 (31.4) | 0.976 (0.662, 1.439) | 0.903 | 0.930 (0.608, 1.421) | 0.737 |
| 23–25 | 62 | 29 (46.8) | 1.871 (1.044, 3.352) | 0.035* | 1.780 (0.945,3.351) | 0.074 |
| **Parents with diabetes or hypertension** | | | | | | |
| No | 250 | 82 (32.8) | Reference | | Reference | |
| Yes | 305 | 103 (33.8) | 1.045 (0.732, 1.490) | 0.809 | 0.894 (0.600,1.331) | 0.581 |
| **Blood Pressure (BP) (mm Hg)** | | | | | | |
| Normal | 453 | 147 (32.5) | Reference | | Reference | |
| Elevated | 39 | 15 (38.5) | 1.301 (0.663, 2.554) | 0.444 | 1.222 (0.588,2.538) | 0.592 |
| High BP Stage 1 | 54 | 22 (40.7) | 1.431 (0.803, 2.549) | 0.224 | 0.887 (0.450, 1.749) | 0.729 |
| High BP Stage 2 | 9 | 1 (11.1) | 0.260 (0.032, 2.100) | 0.206 | 0.145 (0.016, 1.273) | 0.081 |
| **Body Mass Index (kg/m²)** | | | | | | |
| Underweight | 62 | 16 (25.8) | 0.800 (0.431, 1.486) | 0.481 | 0.966 (0.506,1.844) | 0.917 |
| Normal weight | 307 | 93 (30.3) | Reference | | Reference | |
| Overweight | 128 | 47 (36.7) | 1.335 (0.865, 2.061) | 0.192 | 0.832 (0.450,1.539) | 0.558 |
| Obese | 58 | 29 (50.0) | 2.301 (1.302, 4.066) | 0.004* | 0.694 (0.243,1.985) | 0.495 |
| **Body Fat (%)** | | | | | | |
| < 35 | 398 | 115 (28.9) | Reference | | Reference | |
| ≥ 35 | 157 | 70 (44.6) | 1.980 (1.351, 2.901) | <0.001* | 1.266 (0.647, 2.476) | 0.491 |
| **Waist Circumference (cm)** | | | | | | |
| < 80 | 454 | 133 (29.3) | Reference | | Reference | |
| ≥8 0 | 101 | 52 (51.5) | 2.561 (1.650, 3.975) | <0.001* | 1.608 (0.692, 3.737) | 0.269 |
| **Waist-hip Ratio** | | | | | | |
| < 0.85 | 509 | 166 (32.6) | Reference | | Reference | |
| ≥ 0.85 | 46 | 19 (41.3) | 1.454 (0.786, 2.691) | 0.233 | 0.699 (0.329,1.487) | 0.353 |
| **Hemoglobin (g/L)** | | | | | | |
| ≥ 120 | 271 | 91 (33.6) | Reference | | Reference | |
| < 120 | 284 | 94 (33.1) | 0.979 (0.688, 1.393) | 0.904 | 1.014 (0.693,1.483) | 0.942 |
| **Triglyceride (mg/dL)** | | | | | | |
| < 150 | 547 | 178 (32.5) | Reference | | Reference | |
| ≥ 150 | 8 | 7 (87.5) | 14.511 (1.772, 118.847) | 0.013* | 4.399 (0.482,40.153) | 0.189 |
| **Total Cholesterol (mg/dL)** | | | | | | |
| ≤ 150 | 245 | 76 (31.0) | Reference | | Reference | |
| > 150 | 310 | 109 (35.2) | 1.206 (0.844, 1.724) | 0.304 | 1.255 (0.754,2.090) | 0.383 |
| **Low Density Lipoprotein (mg/dL)** | | | | | | |
| ≤ 100 | 382 | 118 (30.9) | Reference | | Reference | |
| > 100 | 173 | 67 (38.7) | 1.414 (0.972, 2.058) | 0.070 | 1.044 (0.626, 1.740) | 0.869 |
| **High Density Lipoprotein (mg/dL)** | | | | | | |
| ≥ 50 | 284 | 70 (24.6) | Reference | | Reference | |
| < 50 | 271 | 115 (42.4) | 2.254 (1.570, 3.236) | <0.001* | 1.980 (1.289,3.042) | 0.002* |
| **High-Sensitivity C-Reactive Protein (mg/L)** | | | | | | |
| < 2.0 | 458 | 126 (27.5) | Reference | | Reference | |
| ≥ 2.0 | 97 | 59 (60.8) | 4.091 (2.592, 6.457) | <0.001* | 2.853 (1.682, 4.838) | <0.001* |

*P <0.05

sensitivity hs-CRP level ($\geq$ 2.0 mg/L), were significantly associated with increased risk of abnormal glycemic status in univariable analyses (odds ratio [OR]: 2.254; 95% CI: 1.570 to 3.236 and [OR]: 4.091; 95% CI: 2.592 to 6.457, respectively; $P < 0.001$) and remained significant in the adjusted analyses (adjusted OR [aOR]: 1.980, 95% CI: 1.289 to 3.042, and [aOR]: 2.853; 95% CI: 1.682 to 4.838, respectively; $P = 0.002$ and $P < 0.001$, respectively) (Table 3).

## 4. Discussion

This study identified a high prevalence of altered glucose metabolism in young Emirati female adults in the UAE. This data confirms the advancing prevalence of diabetic disease in the UAE, particularly given the young age of the study sample. The findings of this study also highlight the complexity of identifying altered glucose metabolism in this population, as many of the common risk factors of hyperglycemia such as obesity, WHR, and family history were shown not to be associated when adjusted for other variables.

The significant prevalence of prediabetes and diabetes, along with the other cardiovascular risk biomarkers has profound implications for healthcare in the UAE. That a young female cohort, with high socioeconomic indicators such as higher education enrollment and metropolitan living would have such a high risk, likely implies a rapid increase in the population more broadly. To help prevent the future toll of cardiovascular disease, action to reduce the prevalence of diabetes in younger people is urgent. Similarly, there is also a high prevalence of overweight and obesity, as well as high WHR and waist circumference in this cohort, particularly in the light of their relative youth. These findings are broadly in line with other studies in the Middle East and North African region [18–20]. It should be noted that most other studies on the prevalence of hyperglycemia, overweight, and obesity focus on a much older population. In the current study the young female cohort approaches the prevalence of larger studies is particularly concerning at a population health level, further adding to the necessity of positive lifestyle intervention in younger individuals in the UAE. This young demographic is an ideal target for education on nutrition, physical activity and other lifestyle changes, as they are less likely to have established disease, and the beneficial effects have decades to improve outcomes. This is particularly relevant in those who are already living with prediabetes and diabetes in youth, as reversing the course of the hyperglycemia, is associated with significant improvements in health outcomes in mid- and later life. Programs aimed at improving nutrition quality and physical activity in the population [21, 22], as well as policy to reduce the carbohydrate and sugar rich foods across industry [23], have been effective in many other nations globally, and should become a priority in the UAE.

The study also showed discordance between HbA1c and FPG in the diagnosis of diabetes when used independently. The two diagnostic criteria showed poor diagnostic agreement, with FPG identifying far fewer participants with elevated HbA1c values identifying far more people with hyperglycemia. However, all of those diagnosed with diabetes using FPG were also identified using HbA1c. This is an area of conjecture, with evidence indicating both under- and overdiagnosis with HbA1c. Our results are in disagreement with those previously reported in a hospital-based study in Korea as the prevalence of diabetes was higher using the FPG criterion (31.6%) compared to the HbA1c criterion (23.5%) [24]. More regionally, a retrospective study in Saudi Arabia found the prevalence of impaired glucose tolerance was 54% when using HbA1c as a diagnostic test, and 60.3% when using FPG as a diagnostic test [25]. In contrast, our results are in agreement with a population-based study among Palestinian Arab population that showed a lower prevalence of diabetes diagnoses using the FPG criterion (4.5%) compared to HbA1c criterion (5.3%) [26]. In a study among the Korean population, the prevalence of diabetes and prediabetes using FPG only (10.5% and 19.3%, respectively) were lower

compared to the prevalence of diabetes and prediabetes using HbA1c as a diagnostic test (12.4% and 38.3%, respectively) [27]. In Canada, using HbA1c only for the diagnoses of overall prediabetes prevalence resulted in a three-fold increase compared to FPG and a six-fold increase among females (FPG 2.22%, HbA1c 13.31%) [28]. Although more participants were diagnosed with diabetes when HbA1c was added as a diagnostic criterion, the simultaneous measurement of FPG and HbA1c (FPG and/or HbA1C) is recommended by the ADA [29] and has been shown to be more sensitive and specific screening tool for identifying high-risk individuals with diabetes and prediabetes at an early stage [30]. This mismatch between HbA1c and glucose has been referred to as a "glycation gap" and it has been observed not only in undiagnosed individuals but also among people with diabetes (type 1 and type 2), and prediabetes [31]. However, the underlying etiology of this mismatch has not yet been elucidated.

In theory, HbA1c and FPG provide different data, as FPG measures blood glucose at a certain point in time, whereas HbA1c reveals long-term glucose control. However, both measures are recommended by the ADA and the WHO for the diagnosis of diabetes, and by the ADA for the diagnosis of prediabetes [5, 7]. Yet, it remains unclear whether one should be preferred over the other. Advantages of using HbA1C include convenience as it does not require fasting, greater preanalytical stability compared to FPG, and minimum day-to-day perturbations during periods of stress and illness. However, it has the disadvantage of being more expensive and influenced by conditions that affect erythrocyte turnover such as abnormal hemoglobin level [32]. Findings of this study showed over half of the studied population (n = 284; 51.2%) had hemoglobin level less than 120 g/l. Studies have shown that iron deficiency anemia shift HbA1c slightly upward but the exact mechanism remains unclear [32]. Therefore, studies recommend correcting iron deficiency before any diagnostic or therapeutic decision is made based on HbA1c [32].

Low HDL level ($<$ 50 mg/dL) and elevated hs-CRP levels ($\geq$ 2.0 mg/L) were independently associated with increased risk of abnormal glycemic status. Hs-CRP is a classic acute-phase inflammatory marker produced in the liver under the stimulation of cytokines such as tumor necrosis factor, interleukin-1 (IL-1), and IL-6 [33]. Previous research linked diabetes with inflammatory responses resulted from β-cell dysfunction [34]. In fact, it was reported that high levels of hs-CRP to be correlated with high levels of HbA1c and FPG [35], and is associated with an increased risk of developing T2D [36]. Similarly, in the general Japanese population elevated hs-CRP concentrations was a significant predictor of diabetes [34].

The factors involved in the relatively poor metabolic health of the UAE and the Middle East and North African (MENA) nations more broadly are many. Rapid westernization and increasing affluence of the population has led to widespread of fast-food uptake, with restaurants serving processed high fat and carbohydrate meals becoming more and more common, particularly in metropolitan regions. This is coupled with decreasing availability of fresh food sources, leading to widespread high-calorie malnutrition, and associated weight gain and hyperglycemia. Additionally, the regional climate makes outdoor physical activity challenging for large portions of the year, and this has led to a lack of exercise culture. This alongside a lack of active encouragement to exercise at a social or governmental level has led to a highly sedentary population. These cultural realities provide simple opportunities for health organizations and governments to intervene to improve the health of the population. Educational campaigns on the benefits of healthy eating and physical activities, subsidization of healthy food choices, and policy to reduce the sugar and calorie content of fast food could provide a strong health improvement in the population.

It is acknowledged that this study has a number of limitations. Firstly, the use of convenience sampling strategy and restricting the sample to female university students. Therefore, the findings may not be extrapolated to all female young adults or non-university population

in the UAE, nor applied to male cohort of the same age. However, our results reflect a high prevalence of diabetes within a small study group, and although it is not a reflection of the whole population, it is very high for female students who are generally more knowledgeable and cautious about health. In addition, studying female students only does not allow for examination of gender differences and factors such as physical activity and stress were not accounted for, which could have had a significant effect on levels of glucose within the body. Secondly, our study did not use the glucose tolerance test, a classified golden standard method which is recommended to be used in conjunction with the FPG, which could have influenced an underestimation of the diagnosing of prediabetes and T2DM. However, unlike the FPG, the HbA1c gives an average level of glucose in the blood for the past couple of months, it would have taken factors such as stress and physical activity into account as it reflects the overall status in the individual for last three months. Thirdly, the cross-sectional design is another limitation of this study, as causal inference cannot be drawn. A study investigating factors influencing diabetes and prediabetes, as well as a larger study group accounting for different demographics is advised to gain a true reflection of the prevalence of T2DM and prediabetes in the UAE.

## 5. Conclusions

This study showed a high prevalence of impaired glucose control amongst young Emirati female university students attending UAEU. This is particularly significant, given that as a population of young females, it would be expected that the prevalence would be comparatively low, signifying that rates of diabetes in the UAE are likely increasing significantly, potentially in excess of projections. As non-communicable diseases, particularly cardiovascular pathology, account for a significant majority of deaths in the nation, measures to control the rise in hyperglycemia are urgently needed to prevent the excessive burden of mortality and morbidity. Public health campaigns should focus on nutrition, physical exercise, and other positive lifestyle interventions, with policymakers regulating food standards, food advertising and packaging labels to assist the population to make better, more informed choices with their diet. Additionally, government and industry should emphasize exercise participation in community-wide campaigns to engage younger people in order to reduce burden of diabetes in the future. Future population studies are needed across the entirety of the country, to monitor the true burden of diabetes in the UAE, and to identify priority targets for intervention to improve the health of the Emirati population.

## Acknowledgments

The authors wish to thank Ms. Rahla Daneshi, Ms. Noura Alneyadi, Ms. Sara Alyousefi, Ms. Abeer Alwahedi, Ms. Moza Alkaabi, Ms. Haneen Ateya and Ms. Asma Numan for their help in recruiting the study participants. We thank all UAEU students who participated in this study.

## Author Contributions

**Conceptualization:** Maysm N. Mohamad, Leila Cheikh Ismail, Lily Stojanovska, Vasso Apostolopoulos, Ayesha S. Al Dhaheri.

**Data curation:** Maysm N. Mohamad.

**Formal analysis:** Maysm N. Mohamad, Jack Feehan, Ayesha S. Al Dhaheri.

**Funding acquisition:** Ayesha S. Al Dhaheri.

**Investigation:** Maysm N. Mohamad, Ayesha S. Al Dhaheri.

**Methodology:** Maysm N. Mohamad, Lily Stojanovska, Amjad H. Jarrar, Ayesha S. Al Dhaheri.

**Project administration:** Maysm N. Mohamad.

**Supervision:** Leila Cheikh Ismail, Lily Stojanovska, Vasso Apostolopoulos, Jack Feehan, Amjad H. Jarrar, Ayesha S. Al Dhaheri.

**Visualization:** Jack Feehan.

**Writing – original draft:** Maysm N. Mohamad, Lily Stojanovska, Amjad H. Jarrar, Ayesha S. Al Dhaheri.

**Writing – review & editing:** Maysm N. Mohamad, Leila Cheikh Ismail, Lily Stojanovska, Vasso Apostolopoulos, Jack Feehan, Amjad H. Jarrar, Ayesha S. Al Dhaheri.

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
