## [Decision Letter · Decision Letter 0]

13 Apr 2021

PONE-D-20-38524

The prevalence of diabetes amongst young Emirati female adults in the United Arab Emirates: A cross-sectional study

PLOS ONE

Dear Dr. Feehan,

Thank you for submitting your manuscript to PLOS ONE. After careful consideration, we feel that it has merit but does not fully meet PLOS ONE’s publication criteria as it currently stands. Therefore, we invite you to submit a revised version of the manuscript that addresses the points raised during the review process.

The reviewers have criticized the methodology and selection criteria of the study. You are invited to revise your manuscript keeping in view that your revision will be re-evaluated by the reviewers. Moreover, when you revise the manuscript, pay special intention to remove all the possible grammatical mistakes and syntax errors.

We look forward to receiving your revised manuscript.

Kind regards,

Muhammad Sajid Hamid Akash

Academic Editor

PLOS ONE

Journal Requirements:

Additional Editor Comments (if provided):

Reviewers' comments:

Reviewer's Responses to Questions

**Comments to the Author**

1. Is the manuscript technically sound, and do the data support the conclusions?

Reviewer #1: Partly

Reviewer #2: Partly

2. Has the statistical analysis been performed appropriately and rigorously? 

Reviewer #1: Yes

Reviewer #2: Yes

3. Have the authors made all data underlying the findings in their manuscript fully available?

Reviewer #1: Yes

Reviewer #2: Yes

4. Is the manuscript presented in an intelligible fashion and written in standard English?

Reviewer #1: No

Reviewer #2: Yes

5. Review Comments to the Author

Reviewer #1: Dear Editor Muhammad Sajid Hamid Akash, PhD

TITLE: The prevalence of diabetes amongst young Emirati female adults in the United

Arab Emirates: A cross-sectional study

Author: Maysm Nezar Mohamad et al.

Comments to the Author

This manuscript nicely presented, an interesting finding that the cross-sectional study of 555 female

college students aged 17-25, enrolled at United Arab Emirates University in Al Ain, UAE. The

Anthropometric analysis, blood pressure, and various biochemical markers were measured using

standard methods. Type 2 Diabetes, impaired fasting plasma glucose (FPG), and elevated HbA1c levels

were examined in the study. Overall, these findings are important and interesting. However, further,

improvement is necessary to solidify the Manuscript.

Here are a few comments and questions:

1. When this study was conducted? Please rationalize the selection of target population. Why the

aged 17– 25 students are selected?

2. How you elaborated the other marker like blood pressure with diabetes or pre-diabetic cases?

3. Why only university students were focused, please add in the study limitation and justify the

sampling biased.

4. What is the major objective of the study? Please focus on it. If its correlation, please explain it

well.

5. Correction needed in writing "SPSS software, version 26.0 (SPSS, Chicago, IL, UASA)".

6. Explain about each test where Univariate and where multivariate logistic regressions was

applied.

Authors address these deficiencies, then the manuscript should be considered for publication.

Reviewer #2: Very good study need some modifications

Please Justify, why the female population of university student was targeted ?

What is the correlation of other parameters with HBa1C ?

Please rationalize the study in introduction?

When this study was conducted ?

Over all alignment of paper is urgently needed

6. PLOS authors have the option to publish the peer review history of their article (what does this mean?). If published, this will include your full peer review and any attached files.

Reviewer #1: **Yes: **DR.MUHAMMAD TARIQ

Reviewer #2: **Yes: **Muhammad Majid Aziz

---

## [Author Response · Author response to Decision Letter 0]

18 May 2021

Comments to the Author

Reviewer 1

This manuscript nicely presented, an interesting finding that the cross-sectional study of 555 female

college students aged 17-25, enrolled at United Arab Emirates University in Al Ain, UAE. The

Anthropometric analysis, blood pressure, and various biochemical markers were measured using

standard methods. Type 2 Diabetes, impaired fasting plasma glucose (FPG), and elevated HbA1c levels

were examined in the study. Overall, these findings are important and interesting. However, further,

improvement is necessary to solidify the Manuscript.

Here are a few comments and questions:

Point 1:

1. When this study was conducted? Please rationalize the selection of target population. Why the

aged 17– 25 students are selected?

Response 1:

Thank you for your kind feedback and valuable comment. The study was conducted between January 2014 and May 2016 as part of a PhD dissertation, yet we believe that the research findings are still relevant and the method of using both HbA1C and FPG is novel and would add value to the current literature in the UAE.

Moreover, the rationale behind selecting the target population has been added to the introduction of the study (line 92-96/ page 4) “focusing on young adults to identify opportunities for early intervention is essential, as there is scope to prevent the transition from prediabetic to diabetic, and potentially reverse the early stage of T2D. Moreover, there is no previous research to date identifying risk factors of altered glucose metabolism using HbA1c and FPG among female young adults.”. In addition, the population structure of the UAE is mainly young adults and was therefore greatly affected by the rapid socioeconomic changes in the country.

Point 2:

2. How you elaborated the other marker like blood pressure with diabetes or pre-diabetic cases?

Response 2:

We thank the reviewer for this valuable observation. Among the study population 11.3% were found to have stage 1 or stage 2 hypertension. However, when logistic regression analysis was conducted (Table 3), no association between high blood pressure and increased risk of abnormal glycemic status was revealed. Moreover, other risk factors for diabetes and cardiovascular disease were discussed in section 3.4 (line 254-277 / page 10-11) Table 3.

Point 3:

3. Why only university students were focused, please add in the study limitation and justify the

sampling biased.

Response 3:

Thank you for your observation. Typically, university students are making the transition from adolescence to adulthood, and during this period they are often forming long-lasting diet and health behaviors that are associated with an increased lifetime risk of type 2 diabetes. Furthermore, to address this concern the limitations section in the discussion has been modified (line 366-369/page 16) “It is acknowledged that this study has a number of limitations. Firstly, the use of convenience sampling strategy and restricting the sample to female university students. Therefore, the findings may not be extrapolated to all female young adults or non-university population in the UAE, nor applied to male cohort of the same age.”

Point 4:

4. What is the major objective of the study? Please focus on it. If its correlation, please explain it

well.

Response 4:

Thank you for your comment. The major objective of this study was to assess the prevalence of prediabetes and T2D among a sample of young Emirati female adults and to identify their risk for altered glucose metabolism. To address this concern, the objectives in the abstract and introduction have been revised and updated accordingly.

Point 5:

5. Correction needed in writing "SPSS software, version 26.0 (SPSS, Chicago, IL, UASA)".

Response 5:

Thank you for your observation. The sentence has been corrected as suggested “USA” (line 195/page 7)

Point 6:

6. Explain about each test where Univariate and where multivariate logistic regressions was

applied.

Response 6:

We thank the reviewer for the constructive comment and agree that it might have been confusing where logistic regression analysis was applied. Therefore, the statistical analyses section (section 2.4) has been updated to include the following paragraph (line 198-201/ page 7) “Univariate and multivariate logistic regressions were used to study the association between age, family history of diabetes or hypertension, blood Pressure, body mass index, body fat, waist circumference, waist-hip ratio, hemoglobin, triglyceride, total Cholesterol, LDL, HDL, High-Sensitivity C-Reactive Protein with abnormal glycemic status as the outcome variable.”. Moreover, the results of the of the logistic regression analysis (Table 3) are presented in section 3.4 (line 254-277 / page 10-11) Table 3.

Authors address these deficiencies, then the manuscript should be considered for publication.

 

Reviewer 2

Very good study need some modifications

Point 1:

Please Justify, why the female population of university student was targeted?

Response 1:

Thank you for your observation. Typically, university students are making the transition from adolescence to adulthood, and during this period they are often forming long-lasting diet and health behaviors that are associated with an increased lifetime risk of type 2 diabetes. Moreover, it is usually very challenging to recruit Emirati young adults and measure their anthropometrics due to cultural restriction.

Additionally, to address this concern the limitations section in the discussion has been modified (line 366-372/page 16)

“It is acknowledged that this study has a number of limitations. Firstly, the use of convenience sampling strategy and restricting the sample to female university students. Therefore, the findings may not be extrapolated to all female young adults or non-university population in the UAE, nor applied to male cohort of the same age.”

“In addition, studying female students only does not allow for examination of gender differences”

Point 2:

What is the correlation of other parameters with HBa1C?

Response 2:

Thank you for this comment. As the ADA recommends the simultaneous measurement of FPG and HbA1c (FPG and/or HbA1C), abnormal glycemic status based on both FPG and HbA1c was used in Table 3 to examine risk factors of diabetes. Logistic regression analysis in Table 3 revealed a correlation between abnormal glycemic status (based on both FPG and HbA1c) and decreased high-density lipoprotein (HDL) level and elevated high-sensitivity C-reactive protein (Hs-CRP) level.

Point 3:

Please rationalize the study in introduction?

Response 3:

Thank you for your comment. Kindly note that the rationale behind selecting the target population has been added to the introduction of the study (line 92-96/ page 4) “focusing on young adults to identify opportunities for early intervention is essential, as there is scope to prevent the transition from prediabetic to diabetic, and potentially reverse the early stage of T2D. Moreover, there is no previous research to date identifying risk factors of altered glucose metabolism using HbA1c and FPG among female young adults.”. In addition, the population structure of the UAE is mainly young adults and was therefore greatly affected by the rapid socioeconomic changes in the country.

Point 4:

When this study was conducted? 

Response 4:

Thank you for your valid point. The study was conducted between January 2014 and May 2016 as part of a PhD dissertation, yet we believe that the research findings are still relevant and the method of using both HbA1C and FPG is novel and would add value to the current literature in the UAE.

---

## [Editor Report · Decision Letter 1]

25 May 2021

The prevalence of diabetes amongst young Emirati female adults in the United Arab Emirates: A cross-sectional study

PONE-D-20-38524R1

Dear Dr. Feehan,

We’re pleased to inform you that your manuscript has been judged scientifically suitable for publication and will be formally accepted for publication once it meets all outstanding technical requirements.

Kind regards,

Muhammad Sajid Hamid Akash

Academic Editor

PLOS ONE
---

## [Editor Report · Acceptance letter]

10 Jun 2021

PONE-D-20-38524R1 

The prevalence of diabetes amongst young Emirati female adults in the United Arab Emirates: A cross-sectional study 

Dear Dr. Feehan:

I'm pleased to inform you that your manuscript has been deemed suitable for publication in PLOS ONE. Congratulations! Your manuscript is now with our production department. 

Kind regards, 

on behalf of

Dr. Muhammad Sajid Hamid Akash 

Academic Editor

PLOS ONE